# Translucent Zirconia in Fixed Prosthodontics—An Integrative Overview

**DOI:** 10.3390/biomedicines11123116

**Published:** 2023-11-22

**Authors:** Andreea Kui, Manuela Manziuc, Adrian Petruțiu, Smaranda Buduru, Anca Labuneț, Marius Negucioiu, Andrea Chisnoiu

**Affiliations:** 1Prosthetic Dentistry and Dental Materials Department, Iuliu Hatieganu University of Medicine and Pharmacy, 32 Clinicilor Street, 400006 Cluj-Napoca, Romania; andreeakui@gmail.com (A.K.); dr.chisnoiu@yahoo.com (A.C.); 2Oral Rehabilitation Department, Iuliu Hatieganu University of Medicine and Pharmacy, 13 Victor Babes Street, 400008 Cluj-Napoca, Romania

**Keywords:** dental materials, translucent zirconia, yttria stabilized tetragonal zirconia, adhesion, accuracy, marginal fit

## Abstract

Over the past two decades, dental ceramics have experienced rapid advances in science and technology, becoming the fastest-growing field of dental materials. This review emphasizes the significant impact of translucent zirconia in fixed prosthodontics, merging aesthetics with strength, and highlights its versatility from single crowns to complex bridgework facilitated by digital manufacturing advancements. The unique light-conducting properties of translucent zirconia offer a natural dental appearance, though with considerations regarding strength trade-offs compared to its traditional, opaque counterpart. The analysis extends to the mechanical attributes of the material, noting its commendable fracture resistance and durability, even under simulated physiological conditions. Various zirconia types (3Y-TZP, 4Y-TZP, 5Y-TZP) display a range of strengths influenced by factors like yttria content and manufacturing processes. The study also explores adhesive strategies, underlining the importance of surface treatments and modern adhesives in achieving long-lasting bonds. In the realm of implant-supported restorations, translucent zirconia stands out for its precision, reliability, and aesthetic adaptability, proving suitable for comprehensive dental restorations. Despite its established benefits, the review calls for ongoing research to further refine the material’s properties and adhesive protocols and to solidify its applicability through long-term clinical evaluations, ensuring its sustainable future in dental restorative applications.

## 1. Introduction

In fixed prosthodontics, there are several types of extra-coronal restorations, including crowns, partial coverage crowns, onlays, and veneers, including implant-supported restorations. Various materials are available for making these reconstructions and can be luted (cemented) conventionally or adhesively, while the most used types of retention for implant-supported prosthesis are screw-retained or cement-retained restorations [1]. While several advantages and disadvantages have been identified and mentioned in the literature regarding retention methods for implant-supported restorations, cement-retained implant restorations seem to be more used due to their aesthetic outcomes [1]. Extra-coronal restorations are still most commonly provided with crowns, but adhesively retained restorations are increasingly popular because they require minimal tooth preparation [2].

In the case of tooth loss, implant-supported restorations have reduced the need for conventionally prepared bridges to some extent; however, tooth-supported fixed partial dentures remain useful options in prosthodontics due to these issues and the status of the remaining dentition [3]. Metal–ceramic FPDs are still perceived as the gold standard for posterior tooth restorations. They provide excellent mechanical properties but lack aesthetic characteristics due to the dark framework underneath, which has to be veneered and can be challenging in areas with insufficient space [4,5,6,7,8,9].

The use of all-ceramic materials has been shown to have superior optical properties, which results in a more tooth-like appearance in terms of color and translucency [8,9,10,11,12,13]. Over the past two decades, dental ceramics have experienced rapid advances in science and technology, becoming the fastest-growing field of dental materials [12,13]. In this same time period, several types of ceramics and processing techniques were developed. They enjoyed increased popularity with the end ceramic system and were given special credit through advances in CAD/CAM [14,15,16].

Throughout history, dental ceramics have been classified in various ways [2,17,18,19]. A more recent classification [2,18] has divided ceramics into two primary categories: (1) glass-based ceramics and (2) polycrystalline ceramics. Zirconia and alumina are the principal polycrystalline compounds used to create high-strength cores, although pure alumina is now used much less because zirconia is much stronger [2]. Zirconia undergoes transformation depending on temperature. At temperatures below 1170 °C, it exists in a form called “monoclinic”, where it resembles a rectangular shape that has been distorted but still has straight sides. Between 1170 °C and 2370 °C, zirconia transforms into a “tetragonal” form, resembling an undistorted rectangular block. At temperatures above 2370 °C, it takes on a “cubic” form, which can be likened to a “Picasso painting” [2].

The tetragonal and cubic phase systems of zirconia become stable at room temperature when solidly dissolved in yttrium (Y), calcium (Ca), magnesium (Mg), cerium (Ce), or other ions with a larger ionic radius than zirconium (Zr) [20,21,22]. The cubic phase is stable at room temperature when over 8 mol% yttria (Y2O3) is added. This is called cubic-stabilized zirconia (CSZ). When yttria is 3 to 8 mol%, tetragonal and cubic phases are mixed at room temperature, and it is called partially stabilized zirconia (PSZ). The tetragonal phases of the zirconia polycrystal (TZP), also known as toughened zirconia, are close to 100% at room temperature when the yttria content is around 3 mol%. In dentistry, this yttria 3-mol% tetragonal zirconia polycrystal (3Y-TZP) is referred to as “white metal” [23,24]. Like yttria, Ce-TZP is tetragonally stabilized with an adequate amount of ceria (CeO2).

Several factors contribute to the failure of fixed partial dentures (FPDs). A cross-sectional study conducted by Chandraik et al. revealed that mechanical factors account for 55.1% of FPD failures, biological factors for 33.3%, and aesthetic reasons for 11.5% [25]. A systematic review published by Tan et al. highlighted a high survival rate of implant-supported FPDs over 5 and 10 years while emphasizing the frequent occurrence of biological and technical complications, necessitating considerable clinician chair time post-procedure [26]. In a retrospective study performed by Dewan et al., several factors contributed to FPDs’ failures, including caries, periodontitis, discomfort, and pain, demonstrating the multifactorial nature of fixed restoration failures [27].

There are still a limited number of clinical studies evaluating the clinical performance and durability of translucent zirconia when used for different fixed prosthetic restorations. Clinical studies with follow-ups of more than five years, clinical randomized trials, and long-term success rate studies are in limited number.

In order to make critical and rigorous decisions regarding oral rehabilitation treatment, professionals need to be able to access scientific evidence. Therefore, the scope of this integrative review is to comprehensively examine the existing literature on the applications of translucent zirconia in the field of fixed prosthodontics (extra-coronal restorations, intra-coronal restorations, fixed partial dentures). We aimed to investigate the latest outcomes regarding some key aspects of these materials, such as mechanical and optical properties, precision in clinical applications, and the quality of the marginal fit and the cementation outcomes. We investigated the latest outcomes regarding some key aspects of high translucency ceramics as compared to established prosthetic materials, therefore including characteristics, such as mechanical and optical properties, surface treatment adhesion, implant-supported features, as well as the quality of the marginal fit and possible advancements in this field.

## 2. Materials and Methods

This literature review was conducted following the PRISMA guidelines (Preferred Reporting Items for Systematic Reviews and Meta-analysis) [28,29]. In addition, the research question was defined through the PICOT format (population, intervention, comparison, outcomes, and time) [30]: P: in vitro case studies including different types of fixed prosthetic restorations (intra-coronal restorations, extra-coronal restorations, fixed partial dentures, implant overdentures); I: using translucent zirconia as the primary material for the fixed prosthetic restorations; C: the use of traditional zirconia or other dental materials commonly used in fixed prosthodontics; O: mechanical strength and wear of restorations made from translucent zirconia, esthetic comparison and adhesion properties, marginal fit accuracy; T: studies and outcomes from the past two decades to provide a contemporary overview regarding the overall performance or translucent zirconia.

### 2.1. Information Sources and Search Strategy

The search was initiated on 1 June 2023 and conducted through 29 July 2023 by two reviewers (AK and MM) using the following bibliographic databases: Medline (PubMed), Scopus, and Embase. For the search, we established 4 search concepts (Table 1), and based on them, we established the keywords and search items, including MeSH terms used to search in all three databases. The exact combination for each search we performed is presented in Table 2. A manual search was also conducted, and references from different studies were included to identify relevant eligible studies.

### 2.2. Eligibility Criteria


*Inclusion criteria*


Randomized controlled trials, cohort studies, case reports, case–control studies—studies involving patients who have undergone fixed prosthodontic treatments using translucent zirconia.In vitro studies that specifically examined translucent zirconia in fixed prosthodontics; in vitro studies using human teeth or relevant analogs.Review articles and meta-analyses that provide comprehensive overviews or evaluations on the topic.Studies specifically discussing ultra-translucent/translucent or highly translucent zirconia.Articles focusing on different types of prosthetic restorations or which evaluated outcomes, such as mechanical strength, aesthetic outcomes, adhesion properties, clinical success rates, longevity, and wear resistance.Studies published in English, completed between 2008 and 2023.


*Exclusion Criteria*


Studies involving the use of traditional or non-translucent zirconia without comparison or relevance to translucent variantsStudies performed on animal subjects only unless they could provide essential and not otherwise available data regarding the use of translucent zirconia.Studies involving the use of hybrid zirconia blocks, multilayered blocks or different types of veneered translucent zirconia specimens.Studies that do not report any of the key outcomes of interest for the review.Articles published in languages other than English.

### 2.3. Data Extraction and Method of Analysis

For data extraction, a standardized form was used and recorded in an Excel table (v.15.17—Microsoft, Redmond, WA, USA). The information extracted included bibliographic information (Authors/title/year of publication/journal), study design and methodology, sample size and demographics, types of translucent zirconia used, clinical indications (what type of prosthetic restoration was investigated), outcomes (mechanical strength, esthetic features, adhesion properties, etc.), number of failures, key findings, and conclusions.

The two reviewers extracted data to ensure consistency and reduce potential bias. Any discrepancies between the two reviewers were resolved through discussion or by consulting a third reviewer (A.P.).

The following step was to evaluate the quality of the articles for this integrative review. While the search included both quantitative and qualitative studies, the evaluation of individual studies requires using different methodologies consistent with the type of search. In order to incorporate every study that aligned with our eligibility criteria and assess their quality, we used the “Mixed Methods Appraisal Tool” [31], which consists of an Excel spreadsheet (v.15.17—Microsoft, Redmond, WA, USA), and is used to accommodate multiple research designs with clearly defined review criteria. The following variables were defined in this investigation: first author’s last name, year of publication, study groups, sample size and thickness, study assessment, and results.

For data evaluation, reviewers developed a template to extract information about the studies (e.g., study purpose, sample, groups, design) and the results that are relevant to the review’s objectives. Data were extracted by the two reviewers and then checked for accuracy and completeness by the other two investigators (A.P. and A.C.).

### 2.4. Scoring Systems Used for Paper Evaluation

In order to include relevant articles based on this search, we have developed a scoring system based on five categories and their corresponding sub-criteria. Each category of the evaluated studies would be assessed with a potential score (Table 3).

## 3. Results

### 3.1. Data Collection

A total of 154 articles were enrolled after applying the search strategy (Table 1 and Table 2). After the elimination of duplicates and eliminating the ones not related to the topic, 121 records were considered for screening. During the first phase, the included articles were selected via their titles/abstracts and their relation to the study question. Therefore, the screening process generated 109 articles, and 97 publications were further assessed for eligibility. Any disagreements were resolved by discussion and by consultation with a fourth one. Finally, a total of 70 publications were included in this review.

The selection process, along with the inclusion decision, is shown in Figure 1, the PRISMA flow diagram.

### 3.2. Description of the Studies and Analysis

Out of the 70 articles included, five studies were other literature or systematic reviews, two of them were clinical studies, and 63 were experimental/in vitro studies. From the total of 70 articles included for the research, 33 articles investigated different aspects regarding mechanical properties and mechanical-optical properties; nine articles investigated the esthetic aspects of translucent zirconia, 14 articles analyzed different adhesion features related to translucent zirconia, seven articles regarding implant-supported restorations and eight articles included were comprehensive insights into zirconia-based prosthetic restorations.

## 4. Discussion

### 4.1. Esthetic Properties of Translucent Zirconia

Alhaream et al. [32] assessed the translucency parameter (TP), contrast ratio (CR) and light blockage percentage of 5Y-TZP and 3Y-PSZ before and after fatigue testing and thermocycling. It found that translucency was inversely related to zirconia thickness and that 5Y-PSZ was more translucent than 3Y-PSZ. However, the differences might not be perceptible to the human eye. Both types were optically stable after the tests. Nevertheless, when comparing the translucency of IPS E.max CAD and high translucency zirconia crowns (LAVA plus^®^), Kanout et al. [33] found that IPS E.max CAD showed significantly higher translucency. Park et al. [34] evaluated the translucency and masking ability of different zirconia types compared to lithium disilicate. It found that while all materials could mask normal dentin shade, they were not capable of masking severely discolored dentin. The study recommends using certain types of zirconia with sufficient thickness to mask titanium. Cho et al. [35] investigated how yttria content influences the translucency and masking ability of zirconia. This study compared the translucency and masking ability of zirconia with different yttria contents. It found that increased yttria content improved translucency but did not adequately mask severely discolored dentin at any thickness.

Mourouzis et al. [36] investigated the effects of milling methods and aging on zirconia’s optical properties. The research assessed the optical properties of zirconia subjected to different milling methods to the milling method (dry or wet milling) and the solution used for milling (fresh distilled water or impregnated water with residues of CAD/CAM ceramic materials) for artificial aging. It concluded that dry milling could result in higher translucency and lower contrast ratio values and that wet milling with impregnated water should be avoided due to saturation of alumina particles. Regarding the effect of mouth rinses on the optical properties of CAD-CAM materials (5Y-TZP zirconia—InCoris TZI, 5Y-TZP zirconia-Zirkonzahn, and lithium disilicate—IPS E.max CAD), Sasany et al. found that color change and translucency reduction were more pronounced in laminate veneer thickness, especially when immersed in certain mouth rinses [37]. Focusing on the impact of the thickness of external stains on zirconia’s optical properties, Lee et al. found that increasing the stain thickness affected lightness, chroma, and hue of 5Y-PSZ with significant changes in optical properties occurring when the stain layer exceeded a certain thickness [38].

Dal Piva et al. compared the staining wear durability of different monolithic ceramics. It found that ceramics with fired staining showed higher durability compared to polymerized ones [39]. Feldspathic ceramic had superior staining durability, followed by zirconia-reinforced lithium silicate and high translucent zirconia. In another study published by Miura et al., the tooth portion and colors of the abutment tooth, along with the resin luting agent, were evaluated in terms of how they affected the final color of zirconia crowns [40]. The results showed that the final color of the monolithic zirconia crowns was significantly influenced by the tooth portion evaluated and the color of the abutment tooth.

Zhang (2019) investigated translucency and slow crack growth and proved that cyclic fatigue degradation was notable yet relatively minor, and that yttria content did not consistently lead to lower strength, with values being equal for 3 mol% and 4 mol% yttria zirconia [41]. Zirconia was significantly more resistant to slow crack growth when compared to lithium disilicate glass–ceramic.

Lee (2022) researched the effect of toothbrushing on the color, translucency and surface roughness of extrinsically stained or glazed 5Y-PSZ [42]. The conclusions stated that significant changes were observed in both shade and translucency parameters between the two types, but there were no significant changes noted after toothbrushing. The surface roughness of the characterized 5Y-PSZ decreased after toothbrushing, while the non-characterized 5Y-PSZ group exhibited an increase.

The effect of shading techniques on fatigue performance and optical properties of 4Y-TZP were investigated by Auzani (2020) [43]. The flexural fatigue strength and the number of cycles required to reach fracture were statistically influenced by pigmentation techniques, with a similar impact on translucency and opalescence. There were no discernible differences in crystalline phase content, topographic pattern, or roughness, although there was an increase in zirconia grain size.

Fouda (2022) found that aging did not significantly affect fracture resistance, and the highest mean fracture load was observed in the case of monolithic zirconia for 4Y and 5Y TZP [44]. However, significant color changes were observed in all groups after the aging process, results in accordance with Da Silva, 2023, but in opposition with Kou, 2019. Fully crystallized lithium disilicate exhibited the highest translucency, and monolithic zirconia for 4Y and 5Y TZP provided the best shade match. Da Silva 2023 showed that different finishing procedures affected roughness, dynamic elastic modulus, microhardness and color [45].

We conclude that there is a nuanced relationship between zirconia’s translucency and its structural components, particularly the yttria content. Higher yttria content enhances translucency but does not guarantee effective masking of severely discolored dentin, necessitating sufficient material thickness or alternative esthetic strategies for certain clinical scenarios. Additionally, the milling methods employed during zirconia fabrication have a pronounced impact on its final optical properties, with dry milling emerging as potentially superior in achieving higher translucency levels.

### 4.2. Mechanical Properties of Translucent Zirconia

#### 4.2.1. Fracture Load, Flexural Strength and Other Mechanical Properties

Abdulmajeed A. (2020) performed tests with or without mastication simulation on 3Y, 4Y, and 5Y-PSZ and concluded that lowering yttria mol% concentration and increasing material thickness significantly increases the fracture load of zirconia [46]. A minimal thickness of 1.2 mm is required for 4Y-PSZ or 5Y-PSZ, and for specimens reduced at 0.7 mm thickness, only 3Y-PSZ survived masticatory simulation. Reducing thickness leads to low fracture resistance for all types of zirconia—Alraheam et al., 2020 [47].

Almansour et al. (2018) investigated the effect of accelerated artificial aging and fatigue on the biaxial flexural strength of 3Y-TZP, 4Y and 5Y-PSZ [48]. The biaxial flexural strength of the high-translucency monolithic zirconia was significantly lower compared with conventional zirconia only without accelerated artificial aging and fatigue, results consistent with Alraheam et al., 2020 [46], and Elsayed et al., 2019 [49]. Aging and fatigue decreased the strength of the zirconia systems tested, but high-translucency monolithic zirconia has a biaxial flexural strength within acceptable clinical values. In opposition to these findings, Jerman (2021) found 3Y, 4Y and 5Y TZP to have the highest flexural strength and translucency after thermo-mechanical aging [50].

Including 3Y-TZP grade standard translucency, 3Y-TZP medium translucency, highly translucent and partially 4Y-TZP, Camposilvan (2018) concluded that a higher proportion of the cubic phase results in improved translucency and stability but comes at the cost of reduced strength and toughness [51]. Glazing does not eliminate the effects of aging, but it does not compromise the material’s strength.

A deeper study of conventional 5Y-PSZ compared with ultra-translucent conducted by de Araújo-Júnior (2022) underlined that sintered 5Y-PSZ has a similar contrast ratio and translucency parameter, Vickers hardness, and fracture toughness with no significant alteration after aging [52]. Irrespective of the processing method, ultra-translucent 5Y-PSZ there is evidence of high aging resistance and translucency stability and strength, corresponding to their application in terms of short-span anterior prostheses.

Flexural strength under monotonic and cyclic load application, hardness, and fracture toughness of different layers of multi-layered zirconia was the subject of a study conducted by Machry (2022). 4-YSZ at the cervical layer showed the highest flexural strength under monotonic and cyclic loads and higher fracture toughness, similar to the transition layer, 4/5-YSZ. Hardness was similar between layers [53].

Yan et al. (2018) found that when adhesively bonded to and supported by dentin, lithium disilicate exhibits similar load-bearing properties to 4Y-PSZ, superior to 5Y-PSZ [54]. Dimitriadis et al. (2022) focused on several mechanical characteristics and discovered that 5Y-PSZ highly translucent zirconia, prepared through the use of CAD-CAM through repeated firing cycles, can be safely used in substructure ceramics for three-unit prostheses involving the molar and substructure ceramics for prostheses involving four or more units, as milling technology is an effective technology [55].

Sahebi et al. (2022) found significantly higher fracture strength in endocrowns [56]. Xu et al. (2015) investigated the effect of test methods and specimen size on flexural strength. This team found that specimens with smaller sizes have higher values than the larger ones and that the edge flaws in ceramic specimens affected tests [57]. Lümkemann N et al., 2021 created 3Y-TZP, 4Y-TZP and 5Y-TZP samples that were subjected to hydrothermal aging in order to study their consequent light transmittance and flexural strength. The findings show that manually colored 4Y-TZP is resistant to this type of aging, as opposed to industrially colored 4Y-TZP [58].

Researchers also focused on crown morphology in a 2023 study conducted by Jurado. They found that crowns without rest seats had a higher fracture resistance than crowns with rest seats; interproximal rest seats were the most resistant of all designs [59].

#### 4.2.2. Surface Treatments

Kim H et al., in 2021, investigated sandblasting to evaluate its effect on high translucency ceramic surfaces [60]. They found that subsurface alterations encompass the appearance of a rhombohedral phase, the presence of micro and macroscopic fractures, as well as the occurrence of both compressive and tensile stresses. Notably, in the case of 3Y-TZP with 110-micrometer particles, the deepest transformed layer exhibits the most substantial compressive stress. As a result, it is suggested that sandblasting particles of 110 µm be employed for 3Y-TZP, while 50 µm particles are recommended for 4Y-PSZ and 5Y-PSZ. Hergeroder et al. found that different surface roughness has no significant differences in flexural strength but cutting the specimens in a fully sintered state reduced flexural strength, especially in 5Y-PSZ air abrasion [61]. Investigating the outcome of different technical tools on the surface temperature and phase composition of crowns, Wertz found that burs had no influence on the phase transformation but caused a shift in preferred orientation, whereas coarse polishers induced a phase transformation to the rhombohedral phase. Fine polishers, on the other hand, did not lead to significant phase transformations or preferred orientation shifts. In comparison to the monoclinic phase, which is linked to low-temperature degradation, the rhombohedral phase has a larger and distorted structure, thereby presenting a higher potential for degradation [62].

In the same context, Kim et al. showed that abrasion of 5Y-PSZ using 110 µm sand resulted in the highest stress value and that using larger particles led to the generation of increased compressive stresses in 3Y-TZP, while 25 µm particles caused residual stresses in 5Y-PSZ. Researchers thus concluded that recommended sandblasting conditions include using 110 µm sand for 3Y-TZP, 90 µm sand for 4Y-PSZ, and 25 µm sand for 5Y-PSZ [60].

In 2022, Alves showed that silica infiltration and polishing-glaze resulted in less volume loss compared to glaze and glass-infiltration techniques, and lithium disilicate exhibited similar roughness when compared to both glazed zirconia materials. Scanning Electron Microscopy (SEM) analysis revealed the removal of the surface treatment following sliding fatigue wear in all materials. Compressive stress was detected on the surfaces of 3Y, while tensile stress was observed on 5Y [63].

#### 4.2.3. Regular or Speed Sintering

Yan M., in 2023, researched the aging resistance of rapidly sintered 5Y-PSZ and concluded that the microstructure, phase composition, and mechanical properties of rapidly sintered 5Y-PSZ materials closely resembled those of conventionally sintered material [64]. Similarly, Jansen (2019) found similar flexural strength for 3Y-TZP in high and regular sintering but a slight decrease in translucency [65]. In 2020, Jerman investigated the effect of high-speed and conventional sintering on the flexural strength of three zirconia materials, both initially and after artificial aging. They found that regardless of the sintering protocols and aging regimens, 3Y-TZP exhibited the highest flexural strength, while 4Y-TZP consistently displayed the lowest flexural strength among the two [66]. The Weibull modulus of the materials subjected to thermo-mechanical aging was adversely affected by high-speed sintering.

A 2020 study conducted by Weidenmann investigated the effects of high-speed sintering, layer thickness, and artificial aging within a chewing simulator on the fracture load and two-body wear of 4Y-TZP crowns. The results demonstrated that high-speed sintering led to reduced two-body wear of the zirconia and yielded fracture load results that were either comparable to or even higher than those of the control group [67]. Crowns manufactured in 4Y-TZP were also investigated by Mayinger in 2022: high-speed sintering with a minimum thickness of 1.0 mm displayed mechanical properties sufficient to withstand masticatory forces, even following a simulated aging period of 5 years [68].

Jeong (2022) concluded that a higher cooling rate did not result in a significant difference in grain size, flexural strength, average transmittance, and translucency, although it did lead to a slight reduction in hardness. The influence of cooling rate during the glazing process on the mechanical and optical properties of 4Y-TZP seems to be minimal and of little clinical significance [69].

Conventionally sintered 4Y-TZP, which was manually colored, exhibited resistance to hydrothermal aging in terms of flexural strength. High-speed sintering prevented color development in manually colored 4Y-TZP but did not influence its resistance to hydrothermal aging, but reverse findings were observed for industrially pre-shaded 4Y-TZP. Lümkemann (2021) [58]. Kim H. (2020) led a study on 3Y-TZP, 4Y and 5Y-PSZ and discovered that rapid cooling was found to enhance translucency due to the formation of a t’-phase with a detrimental effect on the mechanical properties of zirconia. However, the translucency of 5Y-PSZ did not reach the level of lithium disilicate glass–ceramic [70].

#### 4.2.4. Other Mechanical Features

Kim Y, in 2022, proved that the intaglio surface trueness, fracture resistance, and antagonist’s wear volume of the additively manufactured 3Y-TZP crown were found to be clinically acceptable when compared to those of 4Y- or 5Y-PSZ crowns produced via subtractive milling [71]. The two-body wear resistance of 3Y, 4Y, 5Y-TZP when using opposing antagonistic cusps made out of the same material was investigated by De Angelis, 2022. No significant differences in wear among the first-generation 3Y-TZP, second-generation 3Y-TZP, and 4Y-PSZ were found; however, 5Y-PSZ exhibited significantly higher wear compared to the other materials [72].

Liang et al., in 2023, investigated the application of a nanosilica-lithium spray coating on internal and marginal crown surfaces and found no adverse impact on the adaptation of zirconia crowns, thus discovering a clinically viable surface treatment method for zirconia [73].

Schönberger, in 2017, compared the precision of fit of frameworks produced with two different CAD/CAM systems in semi-sintered regular zirconia and high-translucent zirconia blocks. Both systems showed clinically acceptable values but less internal accuracy when regular zirconia was used [74].

Our findings indicate that the fracture load of zirconia is significantly determined by its yttria content and thickness, while lower yttria concentrations and thicker dimensions enhance its durability. However, its strength can be compromised under thermo-mechanical aging, though high-translucency variants tend to retain clinically acceptable strength levels. The material’s cubic phase enhances translucency but may reduce its overall strength, a trade-off that necessitates careful consideration in clinical applications. Additionally, factors such as specimen size, test methodologies, hydrothermal aging conditions, and crown design intricacies significantly influence zirconia’s structural integrity and aesthetic outcomes, pointing to the need for clinical studies regarding the use of translucent zirconia in restorative dentistry.

Regarding surface treatments, studies included in our search indicate that sandblasting, a common surface treatment, induces various subsurface changes, including phase transformations, stress patterns, and microfractures. Notably, different technical tools used for surface modification, like burs and polishers, influence phase composition and structural orientation, with coarse polishers triggering a shift to the rhombohedral phase, which is known for its higher degradation potential. Furthermore, abrasion techniques and particle size selection are crucial for optimizing compressive stresses in zirconia types, enhancing their durability. Investigations also highlight that silica infiltration and polishing-glaze techniques are superior in minimizing volume loss compared to other surface treatments, with implications for the material’s wear resistance. These findings underscore the importance of precise surface treatment protocols to preserve zirconia’s structural integrity and optimize its performance in dental restorations.

### 4.3. Adhesion Features of Translucent Zirconia

Some articles included in this review investigated the bond strength and adhesion techniques. De Angelis et al. found that MDP-based adhesive and self-adhesive resin cements had the highest shear bond strength (SBS) to zirconia, suggesting their suitability for different types of zirconia [75]. Glass ionomer cement showed the lowest SBS. In their research, Ruyter et al. introduced a novel etching technique for zirconia ceramics using low-melting fluoride compounds, achieving good adhesion essential for bonding zirconia restorations [76]. Franco-Tabares et al. investigated the bonding of a 10-MDP-based cement to translucent zirconias, showing promising bonding properties, even after thermocycling [77]. Additionally, Grangeiro et al. explored the effect of multiple firings on the bond strength between translucent zirconia and resin cement, noting an improvement with one to three firings post-sintering [78]. Kim et al. demonstrated that an ethyl–cellulose coating could significantly improve the shear bond strength by preventing saliva contamination on zirconia restorations [79]. Nadal et al. studied the interfacial fracture energy and stress distribution of translucent zirconia and resin cement, highlighting the impact of shear and tensile stresses and the effect of thermal aging [80].

When investigating the influence between surface treatment and material interactions, Packaeser et al. studied the effect of resin cement viscosity on zirconia strength, finding that air abrasion surface treatment enhanced zirconia’s mechanical strength regardless of the resin cement’s viscosity [81]. Mehari et al. evaluated the effects of air abrasion with different materials on zirconia, concluding that aluminum oxide significantly increased bond strength compared to other methods [82]. Khanlar et al. found that air abrasion with different particles and pressures can improve bonding to zirconia, especially when specific combinations of abrasion material and primers are used [83]. Âgren et al. evaluated the shear bond strength of different materials when luted to enamel, noting specific strengths in ZPA compared to WCS [84].

Four included studies investigated the effects of a laser on the adhesion properties. Zhang et al. assessed the transmission of Er:YAG laser energy through zirconia ceramics, noting variations based on ceramic thickness and shade [85]. Birand and Kurtulmus-Yilmaz evaluated the efficacy of Er,Cr:YSGG laser irradiation for debonding zirconia crowns, finding it efficient but cautioning against reusing debonded 5Y-TZP zirconia crowns due to decreased strength [86]. Borba et al. assessed the damage sensitivity of different zirconia types under simulated mouth motion, finding that 5Y-PSZ showed greater strength degradation, indicating its sensitivity to damage [87]. Alammar and Blatz reviewed resin bonding protocols for high-translucent zirconia, confirming the effectiveness of certain protocols and materials for long-term durable resin bonds [88].

In conclusion, research indicates that MDP-based adhesives and self-adhesive resin cements offer superior shear bond strength to zirconia, outperforming other options like glass ionomer cement. Innovative approaches, including a novel etching technique using low-melting fluoride compounds and an ethyl–cellulose coating, have been shown to enhance adhesion significantly. The bond strength is also influenced by various surface treatments, with air abrasion treatments, particularly using aluminum oxide, markedly improving bonding. However, the effectiveness of these techniques can vary based on resin cement viscosity and zirconia type. Laser treatments, especially Er:YAG and Er,Cr:YSGG lasers, have emerged as potent tools for manipulating zirconia surfaces, though their impact can differ based on ceramic properties and application protocols. Notably, while certain high-translucent zirconia systems demonstrate promising bonding properties, they may exhibit sensitivity to damage, underscoring the need for optimized bonding protocols to ensure long-term durability.

### 4.4. Translucent Zirconia Used for Implant-Supported Restorations

Gonzaga et al. (2015) investigated the precision of CAD/CAM systems in creating zirconia and cobalt–chromium frameworks compared to traditional methods. They discovered that CAD/CAM-fabricated frameworks, especially those made of cobalt–chromium, had significantly better fit accuracy than conventionally fabricated ones. Despite the type of fabrication, high levels of passive fit were achieved for all techniques [89]. Cevik et al. reviewed the advancements in CAD-CAM materials for implant-supported dental prostheses. They highlighted the precision and potential of new materials like soft alloys, composite resins, and high-performance polymers. However, they emphasized the need for clinical studies to validate the performance of these materials under various conditions [90].

Zacher et al. conducted an in vitro study comparing the performance and fracture resistance of various materials used in anterior implant-supported prostheses. They found that all tested prostheses survived thermal and mechanical load testing without damage. The fracture forces varied among materials, but the presence of a screw channel did not significantly affect the results. The study concluded that all tested systems were suitable for anterior implant applications [91]. Spitznagel et al. tested the durability and failure modes of various all-ceramic crowns on zirconia implants. They found that all materials tested survived fatigue exposure and showed significant differences in failure loads. Z-HT and Z-ST materials demonstrated the highest reliability, suggesting their suitability for clinical use [92].

Südbeck et al. examined the impact of restoration material and artificial aging on the bending moment of implants with directly screwed restorations [93]. They found that implants without a titanium base showed higher initial bending moments, especially for 4Y-TZP restorations. Artificial aging reduced the bending moment in most subgroups, with no differences found between materials, sintering protocols, or implant types after aging. Biadsee et al. investigated the effect of Titanium-Base Abutment Height revealed that using a 5.5 mm-height ti-base abutment might produce clinically unacceptable color outcomes in certain zirconia crown shades. This finding is crucial for achieving the desired aesthetic results in anterior screw-retained zirconia restorations [94].

### 4.5. Advancements, Applications, and Evaluations of Zirconia-Based Materials in Fixed Prosthodontics

Guncu et al. conducted a long-term clinical study assessing the performance of monolithic zirconia fixed dental prostheses (FDPs) fabricated using digital workflows. The study, which followed 58 patients over six years, found that these FDPs, particularly those with specific connector dimensions, were reliable and demonstrated favorable biological and technical outcomes. No instances of decementation or caries were reported, although some patients showed signs of gingivitis. The study suggests digitally fabricated zirconia FDPs as a viable alternative to traditional restorations [95].

Ozden et al. explored the impact of sintering time on the fit of monolithic zirconia crowns. The in vitro study revealed that short-term sintering protocols significantly affected the marginal fit of 4Y-TZP crowns, though the changes were within clinically acceptable limits. The study emphasizes following manufacturer recommendations for sintering based on the specific zirconia composition [96]. Ghodsi and Jafarian (date not provided) reviewed the properties and applications of translucent zirconia in dentistry. They highlighted the material’s advantages, including less tooth preparation, biocompatibility, and aesthetic superiority over traditional restorations. However, they noted the need for further investigation into the effects of increased translucency on material properties [97].

Arellano Moncayo et al. conducted a narrative review of the modifications in third- and fourth-generation zirconia ceramics, analyzing how these changes affect mechanical and optical properties. The review identified a knowledge gap concerning the specific characteristics of newer zirconia generations, indicating a need for more comprehensive studies [98]. Kongkiatkamon et al. presented an updated review on the classifications of zirconia used in dentistry, discussing the material’s evolution and varied applications. The review serves as a comprehensive guide for professionals, emphasizing the diverse range of zirconia-based restorative materials available in modern dentistry [99].

Schmidt et al. compared the fracture behavior of cantilever FDPs made from different zirconia types. The study found that while all materials were potentially suitable for posterior cantilever FDPs, there were variations in fracture loads influenced by the yttria content and aging procedures. The study calls for more data, especially regarding the use of 5Y-TZP zirconia [100].

Schönberger et al. investigated the precision of fit of zirconia frameworks produced through the use of two different CAD/CAM systems. The study found significant differences in internal fit between the systems, though both were within clinically acceptable ranges. The type of zirconia material used also influenced the fit, with regular zirconia showing higher internal values compared to high-translucent zirconia in certain conditions [74].

Sachs et al. evaluated the fit of full-arch prostheses and single crowns made from translucent zirconia. The study found that single crowns had significantly better accuracy of fit compared to 14-unit FDPs. However, both types of restorations demonstrated clinically acceptable marginal and internal fit, suggesting their suitability for clinical use [101].

### 4.6. Limitations of Our Study

We identified some limitations of this integrative review. Firstly, due to the heterogeneity of study designs, we found it challenging to directly compare the results. In addition, the variability of sample sizes, testing conditions, and aging protocols between the studies included led to an impossibility in terms of performing statistical analysis of the outcomes.

Secondly, the absence of comprehensive clinical studies, particularly in newer zirconia generations, restricts the direct applicability of our findings to real-world clinical scenarios. Additionally, the lack of long-term clinical studies, especially in areas like fixed prosthodontics and implant-supported restorations, raises concerns about the durability and performance of translucent zirconia over extended periods. Another limitation pertains to the lack of standardized protocols across different studies for surface treatments, sintering processes, and bonding procedures, introducing variability in the results and limiting the establishment of universal guidelines. Moreover, the diversity of zirconia types, including 3Y-TZP, 4Y-PSZ, and 5Y-PSZ, with significant variations in properties, may not have been adequately addressed, with the potential of overlooking nuances specific to each type.

## 5. Conclusions

Clinical implications of the data gathered in this overview should help clinicians choose the most suitable type of high translucent zirconia for their specific case, considering esthetic and mechanical properties, but also adhesion on the specific situation to be dealt with.Our research emphasizes the complex nature of translucent zirconia’s aesthetic properties. While it holds promise for high translucency and color adaptability, clinicians and technicians must navigate its nuances and limitations to optimize restorative outcomes. Translucent zirconia presents complex aesthetic properties with the potential for high translucency and color adaptability.Future studies could benefit from a standardized approach to evaluating aesthetic properties, ensuring more consistent results and reliable guidance for clinical application.Despite challenges, translucent zirconia, with its evolving enhancements in esthetic and mechanical properties, stands as a versatile material in restorative and implant dentistry.Ongoing research and technological advancements might contribute to refining the properties of translucent zirconia, expanding its clinical applications.A thorough understanding of its behavior in clinical conditions, meticulous material selection, and adherence to recommended fabrication and treatment protocols are crucial for optimizing the performance and aesthetic outcomes of translucent zirconia.

## Figures and Tables

**Figure 1 biomedicines-11-03116-f001:**
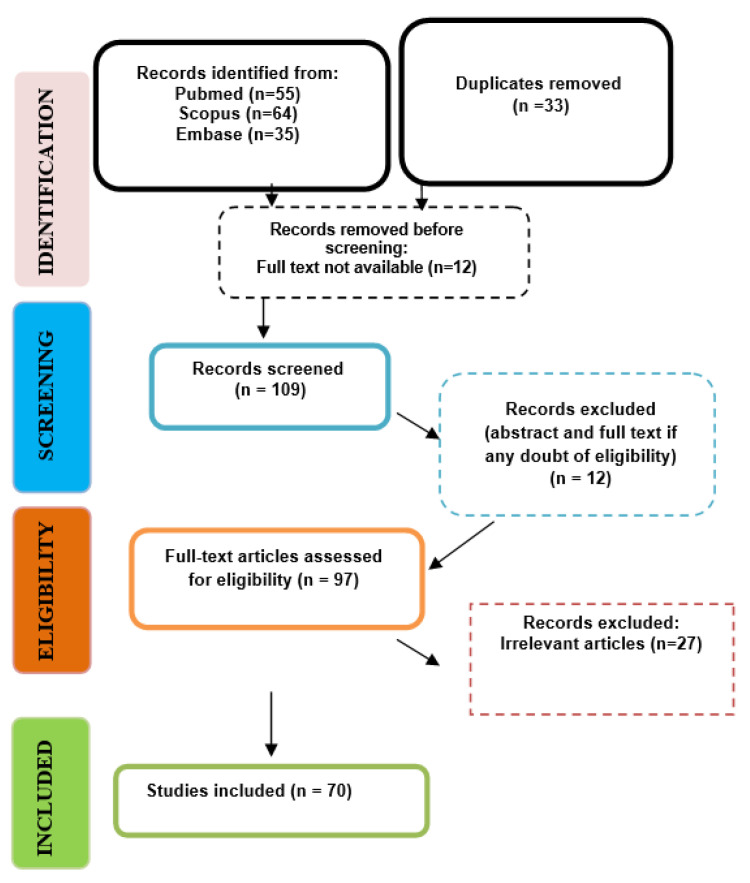
PRISMA flow chart of the selection process.

**Table 1 biomedicines-11-03116-t001:** Concepts used for the literature review.

Concept	Keywords and MeSH Terms
Translucent zirconia	“yttria stabilized tetragonal zirconi*” [Tw] OR “yttria-stabilized zirconi*” [tw] OR “yttria stabilized zirconi*” [tw] OR “YSZ” [tw] OR “high-translucent zirconi*” [tw] OR ultratranslucent zirconi* [tw] OR “5Y-TZP” [tw] OR “6Y-PSZ” [tw]
Prosthetic restorations	“Denture, Partial, Fixed, Resin-Bonded” [tw] OR “Denture, Partial, Fixed” [tw] OR “Crown*” [tw] OR “Dental Veneer*” [tw]
Digital dentistry	“Computer-Aided Design” OR “CAD-CAM”
Accuracy, marginal fit, esthetic outcomes, adhesion and mechanical strength	“accuracy*” [tw] OR “marginal fit” [tw] OR “esthetic outcome*” [tw] OR “adhesion” [tw] OR “mechanical strength” [tw]

**Table 2 biomedicines-11-03116-t002:** Exact combination for each search performed.

Database	Search Terms and Combinations
PubMed	“yttria stabilized tetragonal zirconi*” [Tw] OR “yttria-stabilized zirconi*” [tw] OR “yttria stabilized zirconi*” [tw] OR “YSZ” [tw] OR “high-translucent zirconi*” [tw] OR “ultratranslucent zirconi*” [tw] OR “5Y-TZP” [tw] OR “6Y-PSZ”[tw] “Denture, Partial, Fixed, Resin-Bonded” [tw] OR “Denture, Partial, Fixed” [tw] OR “Crown*” [tw] OR “Dental Veneer*” [tw]“accuracy*”[tw] OR “marginal fit” [tw] OR “esthetic outcome*” [tw] OR “ adhesion” [tw] OR “mechanical strength” [tw] ((“yttria stabilized tetragonal zirconi*” [Tw] OR “yttria-stabilized zirconi*” [tw] OR “yttria stabilized zirconi*” [tw] OR “YSZ” [tw] OR “high-translucent zirconi*” [tw] OR ultratranslucent zirconi*[tw] OR “5Y-TZP” [tw] OR “6Y-PSZ” [tw]) AND (“Denture, Partial, Fixed, Resin-Bonded” [tw] OR “Denture, Partial, Fixed” [tw] OR “Crown*” [tw] OR “Dental Veneer*” [tw])) AND (“accuracy*” [tw] OR “marginal fit” [tw] OR “esthetic outcome*” [tw] OR “ adhesion” [tw] OR “mechanical strength” [tw])
Scopus	“yttria stabilized tetragonal zirconia” OR “yttria-stabilized zirconia” OR “yttria stabilized zirconia” OR “YSZ” OR “high-translucent zirconia” OR “ultratranslucent zirconia” OR “5Y-TZP” OR “6Y-PSZ” “Denture, Partial, Fixed, Resin-Bonded” OR “Denture, Partial, Fixed” OR “Crown*” OR “Dental Veneer*” “accuracy” OR “marginal fit” OR “esthetic outcome” OR “ adhesion” OR “mechanical strength” ((“yttria stabilized tetragonal zirconia” OR “yttria-stabilized zirconi*” OR “yttria stabilized zirconia” OR “YSZ” OR “high-translucent zirconi*” OR “ultratranslucent zirconia” OR “5Y-TZP” OR “6Y-PSZ”) AND (“Denture, Partial, Fixed, Resin-Bonded” OR “Denture, Partial, Fixed” OR “Crown*” OR “Dental Veneer*”) AND (“accuracy” OR “marginal fit” OR “esthetic outcome” OR “ adhesion” OR “mechanical strength”)
Embase	“yttria stabilized tetragonal zirconia” OR “yttria-stabilized zirconia” OR “yttria stabilized zirconia” OR “YSZ” OR “high-translucent zirconia” OR “ultratranslucent zirconia” OR “5Y-TZP” OR “6Y-PSZ” “Denture, Partial, Fixed, Resin-Bonded” OR “Denture, Partial, Fixed” OR “Crown*” OR “Dental Veneer*” “accuracy” OR “marginal fit” OR “esthetic outcome” OR “ adhesion” OR “mechanical strength” ((“yttria stabilized tetragonal zirconia” OR “yttria-stabilized zirconi*” OR “yttria stabilized zirconia” OR “YSZ” OR “high-translucent zirconi*” OR “ultratranslucent zirconia” OR “5Y-TZP” OR “6Y-PSZ”) AND (“Denture, Partial, Fixed, Resin-Bonded” OR “Denture, Partial, Fixed” OR “Crown*” OR “Dental Veneer*”) AND (“accuracy” OR “marginal fit” OR “esthetic outcome” OR “ adhesion” OR “mechanical strength”)

**Table 3 biomedicines-11-03116-t003:** The scoring system used to evaluate the articles included in the research.

Criteria	Sub-Criteria	Score (Points)
Study design and methodology (30 points)	Clear description of study design	5
	Rigorous methodology and appropriate study design for the research question	8
	Adequate sample size justification	5
	Clarity in data collection and analysis methods	7
	Appropriate statistical methods	5
Relevance to research question (20 points)	Direct alignment with the research question or objective	8
	Contribution to the overall objectives of this review	7
Results and findings (25 points)	Clear presentation of results	7
	Thorough analysis of findings	8
	Relevance of results to the study’s objectives	5
	Identification of limitations and potential biases	5
Discussion and conclusion (15 points)	Interpretation of results in the context of the study’s objectives	6
	Thorough discussion of implications	5
	Sound conclusion based on the study’s findings	4
Quality of reporting (10 points)	Clarity and completeness in reporting study details	4
	Adherence to reporting guidelines and standards	3
	Transparency in describing limitations	3
Overall contribution to this research (10 points)	Significance of the study’s findings	4
	Complementary nature to other studies included	3
	Potential impact on informing future research or clinical practice	3

## Data Availability

Data are contained within the article.

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
