# Peer review of "Translucent Zirconia in Fixed Prosthodontics—An Integrative Overview"

_biomedicines, 2023, doi:10.3390/biomedicines11123116_

Round 1
Reviewer 1 Report
Comments and Suggestions for Authors
Manuscript ID: Biomedicines - 2716401. Translucent zirconia in fixed prosthodontics – an integrative overview. Dear authors, it was a pleasure to review your article paper. My comments are mentioned below.
The authors aimed to investigate the latest outcomes regarding some key-aspect of zirconia, such as, mechanical, and optical properties, precision in clinical applications, as well as the quality of the marginal fit and the cementation outcomes. Some comments about the manuscript are described below.
Introduction
1) Insert the meaning of CAD/CAM (row 51).
2) What does 'FPD' signify in row 41? It's only in row 73 that the authors provide the mean for 'FDP.' Please refer to the definition in row 41.
3) use “in vitro” “in situ” in italic.
Materials and Methods
1) Sometimes the authors used a comma after 'et al.' and sometimes they did not 'et al'. Please, standardize the usage of a comma after 'et al.' throughout the manuscript.
2) Did the authors calculate the Kappa score for inter-examiner agreement in the selected articles? If so, please report.
Results
1) The authors included all the selected articles in the discussion section. However, it might be beneficial to create a table that compiles all the data, including author/year (country), sample size, group comparisons, types of analyses (mechanical, optical, surface, etc.), results, conclusions, and other relevant information. Adding this table would enhance the readability of the entire article.
Discussion
1) Row 416: What does mean WCS?
2) The authors have used abbreviations that lack initial explanations. While prosthetic specialists may understand them, other professionals may not be familiar with these acronyms. Please provide the meanings for each acronym the first time they are used in the manuscript.
3) Row 465. There is no need to repeat "fixed dental prostheses" and include "(FDPs)" since the authors have already defined the meaning in the Introduction section.
In general, the present study is interesting, but it requires minor revisions before acceptance.
Author Response
Dear Reviewer #1,
Thank you so much for your detailed review and professional opinion. Please find bellow the answers to your comments and suggestions
Reviewer 1
- Regarding your comments and suggestion: “In the introduction: “Various materials are available for making these reconstructions and can be luted (cemented) conventionally or adhesively. Extra-coronal restorations are still most commonly provided with crowns, but adhesively retained restorations are increasingly popular because they require minimal tooth preparation. [1]” From the authors' point of view, which I fully agree with, the effectiveness of the vein material is amplified or even improved by the use of suitable cement, with pros and cons of both a biological and aesthetic and functional nature. From this point of view I believe what is described in this manuscript is useful, which I believe can enrich the introduction from this point of view:
“Reda R, Zanza A, Cicconetti A, Bhandi S, Guarnieri R, Testarelli L, Di Nardo D. A Systematic Review of Cementation Techniques to Minimize Cement Excess in Cement-Retained Implant Restorations. Methods Protoc. 2022 Jan 17;5(1):9. doi: 10.3390/mps5010009.” – we introduced a new paragraph as follows:
“In fixed prosthodontics, there are several types of extra-coronal restorations, including crowns, partial coverage crowns, onlays, and veneers, including implant-supported restorations. Various materials are available for making these reconstructions and can be luted (cemented) conventionally or adhesively, while the most used types os retention for implant-supported prosthesis are screw-retained or cement retained restorations [1]. While several advantages and disadvantages have been identified and mentioned in the literature regarding retention methods for implant supported restorations, , cement-retained implant restorations seem to be more used due to their aesthetic outcomes [1] “
[1] - Reda R, Zanza A, Cicconetti A, Bhandi S, Guarnieri R, Testarelli L, Di Nardo D. A Systematic Review of Cementation Techniques to Minimize Cement Excess in Cement-Retained Implant Restorations. Methods Protoc. 2022 Jan 17;5(1):9. doi: 10.3390/mps5010009.
- Regarding your suggestion and comments: “The discussion is clear and well written, various topics are covered in a clear and concise manner. I think it would be interesting if the Authors could discuss the absence of statistical analysis for this manuscript, and if it were possible to comment in the discussion also on the exclusion of articles from the inclusion criteria and how these may have limited the results of the review. - we acknowledge that the lack of statistical analysis this as a notable aspect of our study. The nature of our review, focused on integrative analysis of existing literature, inherently involves a wide range of study designs, including randomized controlled trials, cohort studies, case reports, and in vitro studies. Given the diversity of data and outcomes reported in these studies, a meta-analysis or statistical synthesis was challenging and may not have provided meaningful insights due to the heterogeneity of the included studies.
In addition we included 2 more paragraphs I the Discussion section emphasizing the limitations of our study, as follows:
“4.7. Limitations of our study
We identified some limitations of this integrative review. Firstly, due to the heterogeneity of study designs, we found challenging to directly compare results. In addition, the variability of sample sizes, testing conditions, and aging protocols between the studies included lead to an impossibility to perform statistical analysis of the outcomes.
Secondly, the absence of comprehensive clinical studies, particularly in newer zirconia generations, restricts the direct applicability of our findings to real-world clinical scenarios. Additionally, the lack of long-term clinical studies, especially in areas like fixed prosthodontics and implant-supported restorations, raises concerns about the durability and performance of translucent zirconia over extended periods. Another limitation pertains to the lack of standardized protocols across different studies for surface treatments, sintering processes, and bonding procedures, introducing variability in results, and limiting the establishment of universal guidelines. Moreover, the diversity of zirconia types, including 3Y-TZP, 4Y-PSZ, and 5Y-PSZ, with significant variations in properties, may not have been adequately addressed, with a potential of overlooking nuances specific to each type. “
Reviewer 2 Report
Comments and Suggestions for Authors
Dear Authors,
you made a great work! Some improvements are suggested before publication!
The paper is an overview article on the translucent zirconia in fixed prosthodontics.
The Authors made a great work in terms of methodology and the paper sounds scientific and well written.
However, some improvements are mandatory before acceptance.
The abstract is well written, complete and summary in its various aspects. The keywords are complete and appropriate.
In the introduction:
· “Various materials are available for making these reconstructions and can be luted (cemented) conventionally or adhesively. Extra-coronal restorations are still most commonly provided with crowns, but adhesively retained restorations are increasingly popular because they require minimal tooth preparation. [1]” From the authors' point of view, which I fully agree with, the effectiveness of the vein material is amplified or even improved by the use of suitable cement, with pros and cons of both a biological and aesthetic and functional nature. From this point of view I believe what is described in this manuscript is useful, which I believe can enrich the introduction from this point of view:
“Reda R, Zanza A, Cicconetti A, Bhandi S, Guarnieri R, Testarelli L, Di Nardo D. A Systematic Review of Cementation Techniques to Minimize Cement Excess in Cement-Retained Implant Restorations. Methods Protoc. 2022 Jan 17;5(1):9. doi: 10.3390/mps5010009.”
Materials and methods are clear and well explained. The authors did a great job in the explanation of all the variables identified and included in the study. The methodology with which the study was carried out is absolutely clear and repeatable.
Results are easy to understand and comprehensive. All the studied characteristics were reported in tables which are clear and concise.
The discussion is clear and well written, various topics are covered in a clear and concise manner. I think it would be interesting if the Authors could discuss the absence of statistical analysis for this manuscript, and if it were possible to comment in the discussion also on the exclusion of articles from the inclusion criteria and how these may have limited the results of the review.
Conclusions are concise and clear.
Bibliography should be formatted respecting the journal’s requirements, updated, and no improper citations are evidenced.
Figures and labels are clear and easy to comprehend.
English is clear and easy to understand.
Author Response
Dear Reviewer #2,
Thank you so much for the constructive and kind comments and suggestions. The manuscript has been revised according to the suggested modifications.
- Regarding your comments and suggestions: ” There is an issue with the PICO questions and search strategy. In the P (Population) you stated: ''Individuals requiring different types of fixed prosthetic restorations'' while in the search strategy you included in vitro studies that means that no individuals requiring different types of restorations were considered in the study. Also there is an issue when you compare the P (Population) with the O (outcomes) that says: mechanical strength and longevity of restorations made from translucent zirconia/ esthetic outcomes and patient satisfaction/ adhesion properties and marginal fit accuracy, clinical success rate, potential complications, and failure instances. Mechanical strength and marginal fit are mostly evaluated in in vitro study. Therefore, clarify the issue and make the appropriate modifications in the text (please note that this issue alone if not properly addressed may means that the review was not well design since the beginning (with the PICO questions).” you're correct in pointing out the discrepancy between the stated population (P) and the inclusion of in vitro studies in our search strategy. The intention of our study was to provide a comprehensive overview of translucent zirconia, including its in vitro characteristics, which are fundamental to understanding its performance in clinical settings. Insufficient clinical data that could provide standardized data was found, therefore we modified the ‘population’ (P) to “in vitro studies”. Regarding the comparison between the Population (P) and Outcomes (O), you've correctly noted that mechanical strength and marginal fit are frequently evaluated in in vitro studies. This was intentional as our study aimed to provide a comprehensive assessment of translucent zirconia, encompassing both laboratory and clinical aspects. We refined the wording in the Methods section to clarify the aspects you suggested, as follows:
“This literature review was conducted following the PRISMA guidelines (Preferred Reporting Items for Systematic Reviews and Meta-analysis) [28,29]. In addition, the research question was defined through the PICOT format (population, intervention, comparison, outcomes, and time) [30]: P: in vitro case studies including different types of fixed prosthetic restorations (intra-coronal restorations, extra-coronal restorations, fixed partial dentures, implant overdentures), I: using translucent zirconia as the primary material for the fixed prosthetic restorations, C: use of traditional zirconia or other dental materials commonly used in fixed prosthodontics, O: mechanical strength and wear of restorations made from translucent zirconia, esthetic comparison and adhesion properties, marginal fit accuracy, , T: studies and outcomes from the past two decades to provide a contemporary overview regarding the overall performance or translucent zirconia. “
Moher, D.; Liberati, A.; Tetzlaff, J.; Altman, D.G.; PRISMA Group. Preferred reporting items for systematic reviews and meta-analyses: the PRISMA statement. PLoS Med 2009, 6, e1000097. https://doi.org/10.1371/journal.pmed.1000097
Zorzela, L.; Loke, Y.K.; Ioannidis, J.P.; Golder, S.; Santaguida, P.; Altman, D.G.; et al.; PRISMA Harms Group. PRISMA harms checklist: improving harms reporting in systematic reviews. BMJ. 2016, 352, i157. https://doi.org/10.1136/bmj.i157
Hong, Q.N.; Pluye, P.; Fàbregues, S.; Bartlett, G.; Boardman, F.; Cargo, M.; Dagenais, P.; Gagnon, M.-P.; Griffiths, F.; Nicolau, B.; O’Cathain, A.; Rousseau, M.-C.; Vedel, I. Mixed Methods Appraisal Tool (MMAT), version 2018. Registration of Copyright (#1148552), Canadian Intellectual Property Office, Industry Canada.
- Regarding your comments and suggestion: “discuss the possible role of zirconia when considering the antagonist wear in both normal and in a bruxism scenario by adding appropriate references on the topic” as you correctly observed, this topic is of great interest to clinicians. However, there are no studies in a bruxism scenario that could be included in this review, according to the guidelines. Some information on antagonist wear has been provided by Kim Y et al in 2022 (72), De Angelis (73), Alves et al (63), Dal Piva et al (39), as shown in the discussion section.
Reviewer 3 Report
Comments and Suggestions for Authors
The article is interest and provide an up to date overview of the application of translucent zirconia. However, some major corrections must to be done prior for further consideration.
Materials and methods
- There is an issue with the PICO questions and search strategy. In the P (Population) you stated: ''Individuals requiring different types of fixed prosthetic restorations'' while in the search strategy you included in vitro studies that means that no individuals requiring different types of restorations were considered in the study. Also there is an issue when you compare the P (Population) with the O (outcomes) that says: mechanical strength and longevity of restorations made from translucent zirconia/ esthetic outcomes and patient satisfaction/ adhesion properties and marginal fit accuracy, clinical success rate, potential complications, and failure instances. Mechanical strength and marginal fit are mostly evaluated in in vitro study.
Therefore, clarify the issue and make the appropriate modifications in the text (please note that this issue alone if not properly addressed may means that the review was not well design since the beginning (with the PICO questions).
Discussion
- Discuss the possible role of zirconia when considering the antagonist wear in both normal and in a bruxism scenario by adding appropriate references on the topic.
Author Response
Dear Reviewer #3,
Thank you so much for your comments and kind suggestions. The manuscript has been revised according to the suggested modifications.
- Regarding your suggestion “Add a background sentence at the beginning of the abstract”, we introduced a new sentence at the beginning of the abstract as follows: “Over the past two decades, dental ceramics have experienced rapid advances in science and technology, becoming the fastest-growing field of dental materials.”
- Regarding your suggestion “The objectives at the end of introduction section need to be concise and clear.” we introduced a new phrase at the end of the last paragraph in the Introduction section as follows: “We investigated the latest outcomes regarding some key-aspect of high translucency ceramics as compared to established prosthetic materials, therefore including characteristics such as: mechanical, and optical properties, surface treatments adhesion, implant supported features, as well as the quality of the marginal fit and possible advancements in this field.”
- Regarding your comment “In table 2 please add the result of the search and number of studies excluded in each search engine and reasons of exclusion” Table 2 focuses on Exact combination for each search performed, as this information is needed for study reproducibility. Results of search and exclusions are shown in Figure 2, the PRISMA flow diagram and were clarified, as suggested.
- Regarding the question “What was the scoring (evaluation) system used for studies evaluation and filtering. Did authors perform bias analysis.” In our study, a comprehensive scoring system was developed for the evaluation and filtering of studies included in the integrative review. The scoring system was designed to assess various aspects of each study, including study design, methodology, relevance to the research question, presentation of results, discussion, and overall contribution to the integrative review. The total score, calculated based on predefined criteria, provided a quantitative measure of the quality and relevance of each study. To enhance the transparency and reliability of our findings, we implemented this scoring system systematically across all included studies. The scoring system aimed to minimize bias by employing objective criteria to assess the quality and contribution of each study, thereby ensuring a rigorous and standardized evaluation process.
Regarding bias analysis, we performed a thorough examination of potential biases within the individual studies. This involved assessing methodological limitations, sources of bias, and the overall risk of bias inherent in each study. Our approach was guided by established guidelines for bias analysis in systematic reviews, and we explicitly addressed these considerations in the Limitations section of our review. By systematically evaluating and reporting on potential biases, we aimed to provide a clear understanding of the strengths and limitations of the included studies and maintain the integrity of our integrative review.
In addition, we included a paragraph and new table emphasizing the scoring system used for evaluating the articles included, as follows:
“2.4 Scoring systems used for paper evaluation.
In order to include relevant articles based on this search, we have developed a scoring system based on five categories, and their corresponding sub-criteria. Each category of the evaluated studies would be assessed with a potential score (table 3).
Table 3 – the scoring system used to evaluate the articles included in the research.
Criteria |
Sub-criteria |
Score (points) |
Study design and methodology (30 points) |
Clear description of study design |
5 |
Rigorous methodology and appropriate study design for the research question |
8 |
|
Adequate sample size justification |
5 |
|
Clarity in data collection and analysis methods |
7 |
|
Appropriate statistical methods |
5 |
|
Relevance to research question (20 points) |
Direct alignment with the research question or objective |
8 |
Contribution to the overall objectives of this review |
7 |
|
Results and findings (25 points) |
Clear presentation of results |
7 |
Thorough analysis of findings |
8 |
|
Relevance of results to the study’s objectives |
5 |
|
Identification of limitations and potential biases |
5 |
|
Discussion and conclusion (15 points) |
Interpretation of results in the context of the study’s objectives |
6 |
Thorough discussion of implications |
5 |
|
Sound conclusion based on the study’s findings |
4 |
|
Quality of reporting (10 points) |
Clarity and completeness in reporting study details |
4 |
Adherence to reporting guidelines and standards |
3 |
|
Transparency in describing limitations |
3 |
|
Overall contribution to this research (10 points) |
Significance of the study’s findings |
4 |
Complementary nature to other studies included |
3 |
|
Potential impact on informing future research or clinical practice |
3 |
“
- Regarding your comment “Since translucency improved through different generation, please discuss the differences among different generations and how the newest generation differ from the former ones. “ the esthetic characteristic of high translucency zirconia is discussed in section 4.1 section. As discussed through bibliographic entries 31-44, the Yttria content improves esthetic, especially in terms of translucency, but lowers mechanic resistance. Many workflow variants, such as regular or speed sintering, staining technique may influence translucency, thus surpassing generational differences between types. For clarification, a conclusion statement has been added at the end of this section.
- Regarding your comment “Since authors are discussing fracture resistance (subtopic), they should maintain the same mechanical terminology for comparison between different zirconia generations. The authors have involved varying mechanical principles such as: flexural strength, fracture toughness...etc under the same subtopic "fracture resistance". These mechanical principles have different nature, outcomes and implications.” we found beneficial for paper organization the use of subtitles, therefore used several more general terms to enhance review outcomes in the discussion section. The title of this subtopic has been changed to improve adherence to the topics.
- Regarding your suggestion: “Please add indications in which surface treatment of zirconia are needed (polishing, air abrasion...etc) to clarify the need for such procedure in laboratory or clinical settings.” A clarifying paragraph has been added at the end of 4.3 subtopic to further clarify conclusions of 75-88 bibliographic titles, as follows:
“Our findings indicate that the fracture load of zirconia is significantly determined by its yttria content and thickness, while lower yttria concentrations and thicker dimensions enhancing its durability. However, its strength can be compromised under thermo-mechanical aging, though high-translucency variants tend to retain clinically acceptable strength levels. The material's cubic phase enhances translucency but may reduce its overall strength, a trade-off that necessitates careful consideration in clinical applications. Additionally, factors such as specimen size, test methodologies, hydrothermal aging conditions, and crown design intricacies significantly influence zirconia's structural integrity and aesthetic outcomes, pointing to the need for clinical studies regarding the use of translucent zirconia in restorative dentistry.
Regarding surface treatments, studies included in our search indicate that sandblasting, a common surface treatment, induces various subsurface changes, including phase transformations, stress patterns, and microfractures. Notably, different technical tools used for surface modification, like burs and polishers, influence phase composition and structural orientation, with coarse polishers triggering a shift to the rhombohedral phase, known for its higher degradation potential. Furthermore, abrasion techniques and particle size selection are crucial for optimizing compressive stresses in zirconia types, enhancing their durability. Investigations also highlight that silica infiltration and polishing-glaze techniques are superior in minimizing volume loss compared to other surface treatments, with implications for the material's wear resistance. These findings underscore the importance of precise surface treatment protocols to preserve zirconia's structural integrity and optimize its performance in dental restorations.”
- Regarding your suggestion: “Please discuss all mechanical features under one subtopic and keep comparisons per each mechanical feature.”, as there is a large number of mechanical characteristics that authors have studied, these have been grouped as per similarity under several subtopics. Clarifying conclusions have been added on the topic.
- Regarding your suggestion: “Point 4.6. is too long and redundant. Authors may consider re-writing this section or remove it all together. “ we have entirely removed this section, as some information has been summarized and included in the corresponding subtopics.
- Regarding your suggestion: “Please add clinical implications of this review.” clinical implications have been summarized at the end of each subtopic and a paragraph has been added to the conclusions to further underscore this important aspect.
- Regarding your suggestion: “Conclusion can be summarized in bullet points. “we modified the Conclusion section as follows:
“Clinical implications of the data gathered in this overview should help clinicians choose the most suitable type of high translucent zirconia for their specific case; considering esthetic and mechanical properties, but also adhesion on the specific situation to be dealt with.
Our research emphasizes the complex nature of translucent zirconia's aesthetic properties. While it holds promise for high translucency and color adaptability, clinicians and technicians must navigate its nuances and limitations to optimize restorative outcomes. Translucent zirconia presents complex aesthetic properties, with potential for high translucency and color adaptability.
Future studies could benefit from a standardized approach to evaluating aesthetic properties, ensuring more consistent results and reliable guidance for clinical application.
Despite challenges, translucent zirconia, with its evolving enhancements in esthetic and mechanical properties, stands as a versatile material in restorative and implant dentistry.
Ongoing research and technological advancements might contribute to refining the properties of translucent zirconia, expanding its clinical applications.
A thorough understanding of its behavior in clinical conditions, meticulous material selection, and adherence to recommended fabrication and treatment protocols are crucial for optimizing the performance and aesthetic outcomes of translucent zirconia.”
Reviewer 4 Report
Comments and Suggestions for Authors
This paper is genuine and interesting, however the authors should address the following issues to improve the quality of the manuscript:
- Add a background sentence at the beginning of the abstract.
- The objectives at the end of introduction section need to be concise and clear.
- In table 2 please add the result of the search and number of studies excluded in each search engine and reasons of exclusion.
- What was the scoring (evaluation) system used for studies evaluation and filtering. Did authors perform bias analysis.
- Since translucency improved through different generation, please discuss the differences among different generations and how the newest generation differ from the former ones.
- Since authors are discussing fracture resistance (subtopic), they should maintain the same mechanical terminology for comparison between different zirconia generations. The authors have involved varying mechanical principles such as: flexural strength, fracture toughness...etc under the same subtopic "fracture resistance". These mechanical principles have different nature, outcomes and implications.
- Please add indications in which surface treatment of zirconia are needed (polishing, air abrasion...etc) to clarify the need for such procedure in laboratory or clinical settings.
- Please discuss all mechanical features under one subtopic and keep comparisons per each mechanical feature.
- Point 4.6. is too long and redundant. Authors may consider re-writing this section or remove it all together.
- Please add clinical implications of this review.
- Conclusion can be summarized in bullet points.
Author Response
Dear Reviewer #4,
Thank you so much for suggestions. However, we believe that there is a confusion regarding your comments 4 has analyzed as there is no connection between the mentioned aspects and our study.
- No mention of methods for evaluating biocomposites please make sure to do so. Because the topic is really population orientated. (10.1016/j.pnsc.2019.07.004)
- I would like to see a summary table from the team. Which materials are currently used in practice with material characteristics (10.1134/S1087659618060159, 10.1016/j.powtec.2020.04.040)
- The Fourth Industrial Revolution is Klaus Schwab's vision, outlined in his 2016 book of the same name, that the near future will be defined by the widespread adoption of cyber-physical systems in production (Industry 4.0) and the servicing of human needs, including life, work and leisure (Work 4.0).
Reviewer 5 Report
Comments and Suggestions for Authors
This review highlights the significant impact of translucent zirconia in fixed prosthetics, combining aesthetics and durability, as well as its versatility from single crowns to complex bridges, facilitated by advances in digital fabrication.
Сan be adopted with minor modifications
1. No mention of methods for evaluating biocomposites please make sure to do so. Because the topic is really population orientated. (10.1016/j.pnsc.2019.07.004)
2. I would like to see a summary table from the team. Which materials are currently used in practice with material characteristics (10.1134/S1087659618060159, 10.1016/j.powtec.2020.04.040)
3. The Fourth Industrial Revolution is Klaus Schwab's vision, outlined in his 2016 book of the same name, that the near future will be defined by the widespread adoption of cyber-physical systems in production (Industry 4.0) and the servicing of human needs, including life, work and leisure (Work 4.0).
Comments on the Quality of English Language-
Author Response

(The authors gave the same response as above.)

Round 2
Reviewer 3 Report
Comments and Suggestions for Authors
I wasn't able to read the reviewers' answers to my comments because they attached the answers to an other reviewer. However, by checking the text all my comments mainly related to the pico questions were correctly addressed and the text was modified accordingly during the revision process.
Reviewer 4 Report
Comments and Suggestions for Authors
Authors have responded to my comments under "Reviewer 3". All comments were address properly and the paper can be accepted in the current edited form.